# A Structural Comparison of SARS-CoV-2 Main Protease and Animal Coronaviral Main Protease Reveals Species-Specific Ligand Binding and Dimerization Mechanism

**DOI:** 10.3390/ijms23105669

**Published:** 2022-05-18

**Authors:** Chien-Yi Ho, Jia-Xin Yu, Yu-Chuan Wang, Yu-Chuan Lin, Yi-Fang Chiu, Jing-Yan Gao, Shu-Jung Lai, Ming-Jen Chen, Wei-Chien Huang, Ni Tien, Yeh Chen

**Affiliations:** 1Department of Biomedical Imaging and Radiological Science, China Medical University, Taichung 404, Taiwan; samsam172@yaoo.com.tw; 2Division of Family Medicine, China Medical University Hsinchu Hospital, Hsinchu 302, Taiwan; 3Physical Examination Center, China Medical University Hsinchu Hospital, Hsinchu 302, Taiwan; 4Department of Medical Research, China Medical University Hsinchu Hospital, Hsinchu 302, Taiwan; 5AI Innovation Center, China Medical University Hospital, Taichung 404, Taiwan; jiaxin.yu@mail.cmuh.org.tw; 6Institute of New Drug Development, China Medical University, Taichung 404, Taiwan; ycwang322@gmail.com (Y.-C.W.); perry8626@gmail.com (Y.-F.C.); 7Translational Cell Therapy Center, China Medical University Hospital, Taichung 404, Taiwan; xup6m4fm06@gmail.com; 8School of Pharmacy, China Medical University, Taichung 404, Taiwan; u100003056@cmu.edu.tw; 9Graduate Institute of Biomedical Sciences, China Medical University, Taichung 404, Taiwan; sjlai@mail.cmu.edu.tw (S.-J.L.); whuang@mail.cmu.edu.tw (W.-C.H.); 10Research Center for Cancer Biology, China Medical University, Taichung 404, Taiwan; 11Department of Applied Cosmetology, Hungkuang University, Taichung 404, Taiwan; mjchen@sunrise.hk.edu.tw; 12Center for Molecular Medicine, China Medical University Hospital, Taichung 404, Taiwan; 13Drug Development Center, China Medical University, Taichung 404, Taiwan; 14Department of Medical Laboratory Science and Biotechnology, Asia University, Taichung 404, Taiwan; 15Department of Laboratory Medicine, China Medical University Hospital, Taichung 404, Taiwan; 16Department of Medical Laboratory Science and Biotechnology, China Medical University, Taichung 404, Taiwan

**Keywords:** COVID-19, coronavirus, SARS-CoV-2, coronavirus pandemic, viral proteases, crystallography, X-ray, protein conformation

## Abstract

Animal coronaviruses (CoVs) have been identified to be the origin of Severe Acute Respiratory Syndrome (SARS)-CoV, Middle East respiratory syndrome (MERS)-CoV, and probably SARS-CoV-2 that cause severe to fatal diseases in humans. Variations of zoonotic coronaviruses pose potential threats to global human beings. To overcome this problem, we focused on the main protease (M^pro^), which is an evolutionary conserved viral protein among different coronaviruses. The broad-spectrum anti-coronaviral drug, GC376, was repurposed to target canine coronavirus (CCoV), which causes gastrointestinal infections in dogs. We found that GC376 can efficiently block the protease activity of CCoV M^pro^ and can thermodynamically stabilize its folding. The structure of CCoV M^pro^ in complex with GC376 was subsequently determined at 2.75 Å. GC376 reacts with the catalytic residue C144 of CCoV M^pro^ and forms an (R)- or (S)-configuration of hemithioacetal. A structural comparison of CCoV M^pro^ and other animal CoV M^pro^s with SARS-CoV-2 M^pro^ revealed three important structural determinants in a substrate-binding pocket that dictate entry and release of substrates. As compared with the conserved A141 of the S1 site and P188 of the S4 site in animal coronaviral M^pro^s, SARS-CoV-2 M^pro^ contains N142 and Q189 at equivalent positions which are considered to be more catalytically compatible. Furthermore, the conserved loop with residues 46–49 in animal coronaviral M^pro^s has been replaced by a stable α-helix in SARS-CoV-2 M^pro^. In addition, the species-specific dimerization interface also influences the catalytic efficiency of CoV M^pro^s. Conclusively, the structural information of this study provides mechanistic insights into the ligand binding and dimerization of CoV M^pro^s among different species.

## 1. Introduction

Coronaviruses (CoVs) are enveloped RNA viruses that contain positive-sense single-stranded RNA genomes of about 32 kb [1]. Human and animal CoVs generally cause respiratory and enteric diseases [1]. The recent outbreak of coronavirus disease 2019 (COVID-19) caused by Severe Acute Respiratory Syndrome coronavirus 2 (SARS-CoV-2) showed us that CoVs could result in fatal diseases in human and could have huge economic impacts globally [2]. The origin of SARS-CoV-2 is still unclear, but the origin of SARS-CoV [3] and Middle East respiratory syndrome coronavirus (MERS-CoV) [4] have both been identified in bats. Therefore, animals are considered to be natural reservoirs or intermediate hosts for cross-species transmission of CoVs to humans [5]. Coronaviruses belong to the *Coronaviridae* family and can be divided into four genera, Alphacoronavirus (α-CoV), Betacoronavirus (β-CoV), Gammacoronavirus (γ-CoV), and Deltacoronavirus (δ-CoV) [6]. The common coronaviruses that belong to α-CoV include feline infectious peritonitis virus (FIPV) that causes fatal infection in cats, porcine epidemic diarrhea virus (PEDV) that infects neonatal piglets and has caused a severe outbreak in China [7], and canine coronavirus (CCoV) that causes gastrointestinal infection in dogs [8]. The *Tylonycteris* bat CoV HKU4, *Pipistrellus* bat CoV HKU5, and SAR-related civet coronavirus (SARSr-CiCoV), which are closely related to SARS-CoV and MERS-CoV, all belong to β-CoV [9]. Humans can be infected by both α-CoV (human coronavirus (HCoV) 229E and HCoV-NL63) and β-CoV (HCoV-HKU1 and HCoV-OC43) [10]. Chickens are only infected by γ-CoV, such as the infectious bronchitis virus (IBV) [11]. 

Although divergent coronaviruses have evolved to date, the central replication machinery of them is similar. The genomes of coronaviruses usually encode a long polypeptide which must be cleaved by main protease (alternatively named 3-chymotrypsin-like protease, 3CL^pro^) for maturation of several critical components of viral replication machinery. The coronaviral M^pro^ utilizes a catalytic dyad of conserved cysteine and histidine to cleave the peptide bond at the C-terminal side of glutamine residue. The highly conserved catalytic mechanism of coronaviral M^pro^ makes it an attractive therapeutic target for antiviral drug design. In fact, the di-peptidyl bisulfite adduct, GC376, has been successfully applied to treat feline infectious peritonitis in cats [12,13]. Encouraged by this success, GC376 has been widely tested for its in vitro and in vivo efficacy against various coronaviruses, including TGEV (transmissible gastroenteritis virus) [14], FIPV [12], PEDV [15], SARS-CoV [16], MERS-CoV [17], and SARS-CoV-2 [16,18]. In our previous study, we structurally characterized the direct binding of GC376 by SARS-CoV-2 M^pro^, supporting its development as a broad-spectrum antiviral drug [19].

As the closest animals to humans, dogs and cats have the largest potential to be the intermediate hosts to transmit animal or SARS-CoV-related coronaviruses to humans [20]. It has been reported that SARS-CoV-2 RNA was detected in dogs from households of COVID-19 patients in Hong Kong and Italy [21]. Moreover, the SARS-CoV-2 Omicron variant (B.1.1.529) has recently been detected in cats and dogs living with COVID-19 patients in Spain [22]. Another concern is the potential threats of animal CoV variations which may cause the next global pandemic. The novel CCoV strain, which harbors sequence variations in the spike gene, was identified in eight hospitalized patients with pneumonia in Malaysia [23]. Thus, surveillance of domestic animals living with humans is important for public health.

To better understand the similarities and differences between human and animal CoVs, we resolved the crystal structure of CCoV M^pro^ in complex with the broad-spectrum anti-coronaviral drug, GC376. The structural comparison between the CCoV M^pro^_GC376 and SARS-CoV-2 M^pro^_GC376 complexes revealed several distinct structural features that differentiated both of them. The structural information provided here should be very helpful for anti-coronaviral drug design to prevent or treat coronavirus-causing diseases in dogs.

## 2. Results

### 2.1. GC376 Is a Potential Lead Compound against CCoV M^pro^

The Medical Subject Headings (MeSH) entry term for GC376 is “sodium (2S)-2-((S)-2-(((benzyloxy)carbonyl)amino)-4-methylpentanamido)-1-hydroxy-3-(2-oxopyrrolidin-3-yl)propane-1-sulfonate”. GC376 contains γ-lactam glutamine surrogate at the P1 position, leucine at the P2 position, a protecting benzyloxycarbonyl group at the P3 position, and a bisulfite adduct, proposed to be converted into aldehyde form by leaving the sodium bisulfite [14] (Figure 1A). Since there is little information regarding the inhibitory effect of GC376 against CCoV M^pro^, biophysical and biochemical assays were conducted. The results showed that GC376 could effectively block the protease activity of CCoV M^pro^ in FRET-based assay in vitro (Figure 1B). The binding of GC376 thermodynamically stabilized the conformation of CCoV M^pro^ by increasing the melting temperature (*T*_m_) by about 7.8 °C at a concentration of 30 μM (Figure 1C). Together, these data suggest that GC376 stabilizes the dimer formation of CCoV M^pro^ by binding to the substrate-binding site and inactivating the catalytic cysteine residue by a covalent inhibition mechanism, similar to previously reported.

### 2.2. Overall Structure of CCoV M^pro^ in Complex with GC376

To further validate the inhibitory effect of GC376 on CCoV M^pro^ at the atomic level, we determined the complex structure of CCoV M^pro^_GC376 to be a 2.75 Å resolution structure in the *C*2 space group (Table 1). The asymmetric unit (A.U.) contains eight protomers of CCoV M^pro^, which assemble into four functional dimers (Appendix A). Dimerization is critical for M^pro^’s protease activity, because the N-terminal residue Ser1 of one protomer is part of the substrate-binding pocket of the other protomer (Figure 2A). Similar to other coronaviral M^pro^, CCoV M^pro^ can be divided into three domains: domain I (residues 11–100), domain II (residues 101–198), and domain III (residues 199–299) (Figure 2B). Domains I and II adopt conserved chymotrypsin-like folds, in which the GC376 binds into the cleft between them (Figure 2B). Domain III is an alpha-helical domain, which mainly mediates the dimerization (Figure 2A,B). The root mean square deviations of the eight protomers of CCoV M^pro^ range from 0.321 to ~0.474 Å for 231~268 Cα atom pairs, showing nearly identical conformations among them (Figure 2C and Appendix A). However, the GC376 bound to each protomer shows obvious structural variations, especially the P3 protecting group (Figure 2D), suggesting that the malleability of the substrate-binding pocket of CCoV M^pro^ enables antiviral drug design.

### 2.3. GC376 Covalently Linked to Catalytic Cys144 of CCoV M^pro^, Forming an (R)- or (S)-Hemithioacetal

As shown in Figure 1A, the aldehyde warhead of GC376 can react with the thiol group of the catalytic cysteine of M^pro^ in two different ways, forming hemithioacetal in an (R)- or (S)-configuration (Figure 3A). In a previous study [19], we resolved the SARS-CoV-2 M^pro^ complexed with G376, which formed both (R)- and (S)-configurations in the same substrate-binding pocket (Figure 3B). By contrast, the current available structures demonstrated that GC376 formed only a (S)-configuration with animal coronaviral M^pro^, including PEDV M^pro^ [15] and TGEV M^pro^ [14] (Figure 3C). Interestingly, the CCoV M^pro^_GC376 complex structure resolved in this study showed five GC376s in the (R)-configuration and three GC376s in the (S)-configuration in the active sites of CCoV M^pro^ (Figure 3D,E and Appendix A). Similar to other M^pro^-GC376 structures, the hydroxyl group of the (R)-hemithioacetal of GC376 forms a hydrogen bond with the imidazole ring of H41 of CCoV M^pro^ (Figure 3D). However, the hydroxyl group of (S)-hemithioacetal of GC376 forms weak H-bonds with the backbone amide of G142 and C144 (3.9 Å and 4.0 Å, respectively) of CCoV M^pro^ (Figure 3E).

### 2.4. Three Conserved Structural Features Dictate the Substrate-Binding Pocket of CoV M^pro^s

To gain more insights into the similarities and differences between SARS-CoV-2 M^pro^ and animal CoV M^pro^s, structural and bioinformatic analyses of representatives of CoV M^pro^s were conducted (Figure 4A–D, Appendix A). Next, we discuss the structure–activity relationship (SAR), sequence conservation, and evolutionary relationship between them. First, the nearly invariant specificity towards the glutamine residue at the P1 position of the M^pro^’s peptide substrates can be seen by the highly conserved residues constituting the S1 subsite. Four identical residues, i.e., F140, H163, E166, and H172 in SARS-CoV-2 M^pro^ are shared among all the aligned CoV M^pro^s (Appendix A). L141 in SARS-CoV-2 M^pro^ can be replaced by Ile in some CoV M^pro^s (Appendix A). The significant difference lies in the N142 residue in SARS-CoV-2 M^pro^, which is substituted for Cys in M^pro^ from MERS-CoV, bat CoV, HCoV-HKU1, and HCoV-OC43, and for Ala in M^pro^ from IBV, FIPV, TGEV, and CCoV (Appendix A). Consistently, complex structures of M^pro^_GC376 show that the P1-γ lactam ring of GC376 forms three H-bonds with conserved F139/H162/E165 among M^pro^ from TGEV, PEDV, CCoV, and F140/H163/E166 in SARS-CoV-2 M^pro^ (Figure 3B–E). The sidechain of N142 protrudes into the substrate-binding pocket of SARS-CoV-2 M^pro^ and makes additional contacts with P1-γ lactam ring and P3 protecting group of GC376 (Figure 4D) in contrast with the fewer contacts by A141 in CCoV M^pro^ and TGEV M^pro^ (Figure 4D). Second, H41, M49, Y54, M165, and D187 consist of the hydrophobic S2 subsite of SARS-CoV-2 M^pro^. H41 and D187 are invariant among all CoV M^pro^s (Appendix A). Y54 can be replaced by Trp in IBV M^pro^, while M165 can be substituted for Leu in PEDV M^pro^ and HCoV-NL63 M^pro^ (Appendix A). A striking difference that distinguishes M^pro^s from alpha- and beta-coronaviruses was identified: the conserved M49/L49 in β-CoV M^pro^ is absent in all α-CoV M^pro^s and is replaced by a conserved sequence motif (^45^-SXTT-^48^) (Appendix A). The resolved CoV M^pro^ structures show that the sequence motif forms loop conformations in TGEV M^pro^, PEDV M^pro^, FIPV M^pro^, and CCoV M^pro^, in contrast with the α-helix formed in SARS-CoV-2 M^pro^ (Figure 4A,B). This loop (residues 45–48) is relatively flexible as revealed by PEDV M^pro^, which moves away from the substrate-binding pocket (Figure 4B). The conserved T47 in α-CoV M^pro^ could mediate indirect H bonding in water molecules to interact with GC376 (Figure 3C). By contrast, the structurally equivalent M49 in SARS-CoV-2 M^pro^ makes more hydrophobic interactions with the P2-Leu residue of GC376 (Figure 4B,D and Figure 5A). Third, the L167 and Q192 residues that participate in the formation of the S4 subsite of SARS-CoV-2 M^pro^ are invariant among all CoV M^pro^s (Appendix A). F185 can be replaced by Tyr in some CoV M^pro^s (Appendix A). The second evolutionary conserved feature identified in this region is the Q189 in β-CoV M^pro^s, which is replaced by P188 in α-CoV M^pro^s (Figure 4B,D, Appendix A). As revealed by the GC376 bound M^pro^ structures, Q189 of SARS-CoV-2 can directly make one or two H-bonds with the backbone of GC376, instead of indirect hydrogen bonding by T47 from TGEV M^pro^ (Figure 3B,C). In summary, three distinct sequences and structural features in the S1, S2, and S4 subsites, together differentiate their interactions with substrates among different species of CoV M^pro^s. Furthermore, the three unique features in SARS-CoV-2 M^pro^ (M49, N142, and Q189) narrow the entrance of the substrate-binding pockets (7.1 Å, Figure 5A) as compared with the entrance of PEDV M^pro^ (10.4 Å, Figure 5B), TGEV M^pro^ (14.0 Å, Figure 5C), and that of CCoV M^pro^ (13.3 Å, Figure 5D), which probably affect the substrate entry and the catalytic efficiency of CoV M^pro^s.

### 2.5. Species-Specific Dimerization of CoV M^pro^s

In addition to the substrate-binding residues, dimerization is another critical determinant for M^pro^’s activity. It has been reported that devoid of domain III, SARS-CoV M^pro^ forms only a monomeric form and is nearly inactive [24]. Therefore, targeting dimerization of CoV M^pro^ could be an effective approach for anti-coronaviral drug design. Indeed, a recent study identified several lead compounds that targeted two allosteric sites other than the substrate-binding pocket of SARS-CoV-2 M^pro^, which blocked viral replication of SARS-CoV-2 in a cell-based assay [25]. Here, we structurally compare the important residues constituting the dimerization interface of both CCoV M^pro^ and SARS-CoV-2 M^pro^. SARS-CoV-2 M^pro^ forms a tight dimer [26] with a buried interface area of about 1400 Å^2^, but the dimerization interface of CCoV M^pro^ resolved in this study were calculated to be ~1200 Å^2^ among the four non-crystallographic dimers in the asymmetric unit (Appendix A). Two important factors contribute to the dimerization interface of CoV M^pro^: the N-finger of one protomer squeezing between domain II and III of the other protomer and the dimerization between domain III. We found that there was a large gap between domain III of the two protomers of CCoV M^pro^ (distance between the Cα atoms of G281 was 7.3 Å, Figure 6A). By contrast, the two protomers of SARS-CoV-2 M^pro^ were relatively closer to each other (distance between the Cα atoms of A285 was 5.3 Å, Figure 6B). The three residues, i.e., S284, A285, and L286, of each protomer of SARS-CoV-2 M^pro^ together form a hydrophobic core at the interface, while the equivalent residues are separated by a long distance in the CCoV M^pro^ dimer (Figure 6A,B). Tight dimer packing against domain III of each protomer is positively correlated with better catalytic efficiency as revealed by the 3.6-fold enhancement of protease activity of SARS-CoV M^pro^ carrying S284-T285-I286/A mutations [27]. Furthermore, two salt bridges (R4-E290) formed between domain III of the two protomers of SARS-CoV-2 M^pro^, which could strengthen dimerization, are absent in CCoV M^pro^ dimers (Figure 6A,B). Instead, the equivalent R4 residues in CCoV M^pro^ form hydrogen bond interactions with the main chain of G126 (Figure 6A,B). In conclusion, the decreasing protomer–protomer interactions mediated by domain III of CCoV M^pro^, as compared with SARS-CoV-2 M^pro^, probably decrease its catalytic activity, suggesting different adaptions of CoV M^pro^s in different species under evolutionary pressure.

## 3. Discussion

In this study, we resolved the first crystal structure of GC376 bound CCoV M^pro^ and structurally compared it with other GC376_M^pro^ structures. We identified three structural features that distinguish SARS-CoV-2 M^pro^ as a stronger ligand binder than TGEV M^pro^, PDEV M^pro^, and CCoV M^pro^. Interestingly, these three critical residues M49, N142, and Q189 have previously been proposed to be gate-regulated switches for regulation of substrate binding by SARS-CoV M^pro^ [28], indicating their importance in ligand recognition. Through extensive bioinformatic analysis, we found amazing correlations among evolutionary relationships of CoV M^pro^s and two of the three structural features (Appendix A). First, the β-CoV M^pro^s contain M or L residue within the S2 subsite, while α-CoV M^pro^s contain T residue at an equivalent position (Appendix A). Second, β-CoV M^pro^s contain conserved Q residue within the S4 subsite, while α-CoV M^pro^s contain P residue at an equivalent position. A more divergent IBV M^pro^ from γ-CoV contains a negatively charged E residue in the same position (Appendix A). The structural, functional and evolutionary relationships among different CoVs and CoV M^pro^s may reflect their adaptions to different host species. For example, it has been found that γ-CoVs only infected avian species, while α-CoVs and β-CoVs mainly infected mammals. However, the recombination of coronaviruses may blur the boundaries and cause cross-species transmission.

The structural flexibility and plasticity of the substrate-binding pocket of SARS-CoV-2 have been widely investigated [29,30,31]. By contrast, these characteristics have seldom been explored in animal CoV M^pro^s due to a lack of structural information. Here, we resolved the structures of eight different protomers of CCoV M^pro^ in one asymmetric unit. Although the structural variations among them are small, the eight GC376s bound in the substrate-binding pockets exhibit different conformations, suggesting that the malleability of substrate-binding pocket of CCoV M^pro^ could accommodate a certain degree of variations in substrates or drugs. Thus, the complex structures of CCoV M^pro^_GC376 resolved in this study provide the first example of the plasticity of the substrate-binding pocket of animal CoV M^pro^ and could serve as a good starting point for further structure-guided drug design.

Dimerization of M^pro^/3CL^pro^ mediated by domain III is unique to viruses from the *Coronaviridae* family, but absent in viral 3C^pro^ or 3CL^pro^ in the *Picornaviridae* and *Caliciviridae* families, which are made up of only domain I and II [14]. Dimerization is important for the catalytic activity of CoV M^pro^ by stabilizing the conformation of the oxyanion loop [32]. To gain more insights into the mechanism underlying the formation of dimeric CoV M^pro^s, we analyzed several published structures of CoV M^pro^s and found that most of them form an extensive dimerization interface of ~1300–1400 Å^2^. Some SARS-CoV M^pro^ and FIPV M^pro^ (PDB: 4ZRO) showed reduced dimerization interface which was similar with that of CCoV M^pro^. Dimerization interfaces can be influenced by many factors, such as extra residues at N-terminus, structural flexibility, crystallization conditions, and different bound ligands. In addition, we found that SARS-CoV-2 M^pro^ harbored the largest number of H-bonds and an additional ionic pair (R4-E290) at the dimerization interface, which could partially account for its high catalytic efficiency. The R4 residues are replaced by V4 residues in HCoV-HKU1 M^pro^, HCoV-OC43 M^pro^, MERS-CoV M^pro^, and bat CoV HKU4 M^pro^ (Appendix A). The R4 residues of other CoV M^pro^s, such as PEDV M^pro^ (PDB: 6L70), HCoV-229E M^pro^ (PDB: 2ZU2), and HCoV-NL63 M^pro^ (PDB: 3LTO), form H-bond interactions with the main chain of G126 similar to that of CCoV M^pro^. In summary, the structure–activity relationships of these evolutionary conserved structural features of CoV M^pro^s deserve to be further explored in future studies. Altogether, the species-specific differences in three sites of substrate-binding pocket and two sites of dimerization interface could be important structural epitopes for specific monoclonal antibody development or specific antiviral drug design.

## 4. Materials and Methods

### 4.1. Cloning, Expression, and Purification of CCoV M^pro^

The gene fragment of full-length main protease from canine coronavirus (CCoV M^pro^, UniProtKB: P0C6F7.1, a.a. 3299–3604) was *E. coli* codon optimized, chemically synthesized, and then subcloned into pSol SUMO vector for generation of N-terminal His_6_-SUMO tagged CCoV M^pro^. The constructed plasmid was transformed into *E. coli* BL21(DE3) competent cells and cultured overnight in an LB agar plate containing 50 μg/mL kanamycin. The colony containing the desired plasmid was confirmed by sequencing. For large-scale expression, the overnight bacterial cultures carrying the plasmid pSol SUMO_CCoV M^pro^ was diluted 1:100 with fresh LB medium and cultivated continuously at 37 °C with shaking at 200 rpm until the OD_600_ reached 0.6–0.8. The induction of protein expression was carried out by the addition of a final concentration of 0.2% L-rhamnose and continued incubation at 20 °C for 24 h. The bacterial cultures were harvested by centrifugation at 4 °C, 6000 rpm for 30 min. The cell pellets were resuspended in lysis buffer containing 20 mM Tris 8.0, 500 mM NaCl, 10% glycerol, 5 mM TCEP, and 5 mM imidazole and lysed by sonication on ice. The supernatant containing His_6_-SUMO tagged CCoV M^pro^ was separated from cell debris by centrifugation at 4 °C, 18,000 rpm for 30 min, and then loaded into a 5 mL HisTrap™ HP column (Cytiva, Marlborough, MA, USA) for affinity purification by an ÄKTA go protein purification system (Cytiva, Marlborough, MA, USA). The target proteins were eluted by lysis buffer containing a stepwise gradient of imidazole (20, 40, 100, and 200 mM), and then pooled together for further TEV cleavage to remove the His_6_-SUMO tag. After TEV cleavage, the CCoV M^pro^ containing an additional Gly residue at the N-terminus was separated from His_6_-SUMO tag again using an HisTrap™ HP column. The purified CCoV M^pro^ was further applied to size-exclusion chromatography using a Superdex 200 10/300 GL column (Cytiva, Marlborough, MA, USA). The fractions containing active dimeric CCoV M^pro^ were concentrated using an Amicon Ultra-15 centrifugal filter unit (Merck Millipore, Burlington, MA, USA) to the final concentration of 33.2 mg/mL in storage buffer (20 mM Tris 8.0, 200 mM NaCl, 5% glycerol, and 1 mM TCEP). The purified proteins were stored at −80 °C until use.

### 4.2. Fluorescence Resonance Energy Transfer (FRET)-Based Assay

The inhibitory effect of GC376 on CCoV M^pro^ was rapidly determined using a FRET-based assay [19]. Briefly, 9.4 μM CCoV M^pro^ was incubated with 120 μM GC376 in the assay buffer containing 20 mM Tris 7.8 and 80 mM NaCl, at room temperature for 30 min. Then, the fluorescent substrate (final concentration of 20 μM) was added to start the proteolytic reaction. The change in fluorescence (Ex/Em = 434 nm/474 nm) was monitored using a Synergy H1 microplate reader (BioTek, Winooski, VT, USA) for one hour at room temperature. The data points from the first 900 s were used to calculate the initial velocity (V_0_) and normalized to a DMSO control. The experiments were performed in three replicates.

### 4.3. Protein Thermal Shift Assay

The thermal stability of CCoV M^pro^ was accessed by a previously established method [33]. Briefly, 7.5 μM CCoV M^pro^ was incubated with 7.5, 15, 30, 60, or 120 μM GC376 in the assay buffer containing 25 mM Tris pH 8.0, 150 mM NaCl, and 5X SYPRO Orange dye (Sigma-Aldrich, Burlington, MA, USA), at room temperature for 30 min. Then, the thermal shift assay was conducted on a CFX96 RT-PCR instrument (Bio-Rad, Hercules, CA, USA) with a temperature gradient from 25 to 85 °C in 0.3 °C increments of 12 s intervals.

### 4.4. Crystallization and Structural Determination of CCoV M^pro^ in Complex with GC376

First, the optimal protein concentration of CCoV M^pro^ for crystallization was determined using a pre-crystallization test (Hampton Research, Aliso Viejo, CA, USA). Two-fold molar excess of GC376 was incubated with purified CCoV M^pro^ at 4 °C for 1 h, and then subjected to crystallization condition screening. The initial crystallization hit was identified in a mother liquid containing 0.2 M sodium citrate, 0.1 M Bis Tris propane 7.5, and 20% *w*/*v* polyethylene glycol (PEG) 3350 at 20 °C. Manual adjustment of the crystallization condition was applied to improve the quality of protein crystal and the best condition was obtained with a mother liquid containing 0.2 M sodium citrate, 0.1 M Bis Tris propane 7.0, and 25% *w*/*v* PEG 3350 at 20 °C. The mother liquid containing additional 20–25% glycerol was used for cryoprotection under liquid nitrogen. The X-ray diffraction data of the native CCoV M^pro^_GC376 complex were collected, indexed, integrated, and scaled using the HKL2000 software [34] from the beamline TPS 05A at the National Synchrotron Radiation Research Center (NSRRC) in Taiwan. Then, the phase problem was solved by the molecular replacement (MR) method using the Molrep program [35]. The MTZ files were obtained by using the Scalepack2mtz program embedded in the CCP4 interface (version 7.1.015) [36]. The solvent content analysis was conducting using the Matthews_coef program in the CCP4 interface [37]. Subsequent refinement was carried out by using the Refmac5 program [38]. A detailed structural adjustment was performed using the “real space refine zone” function of the COOT software (version 0.9.4) [39]. Water molecules were added to the models using the “find waters” function of COOT with the density between a 1.0 and 2.0 sigma cutoff. Potential ligand binding sites were identified by using the “find unmodelled blobs of density” function. The structural models of ligands were generated using the eLBOW program in the Phenix software (version 1.13-2998) [40]. The quality of the data was checked using the validation server of RCSB PDB (https://www.rcsb.org/ accessed on 18 April 2022).

### 4.5. Bioinformatic Analysis

Multiple sequence alignment was performed using Clustal Omega [41]. The phylogenetic tree analysis was performed using W-IQ-TREE [42].

## 5. Conclusions

The broad-spectrum anti-coronaviral drug GC376 can bind to the substrate-binding pocket of CCoV M^pro^ and can inhibit its protease activity by covalently linking to the catalytic residue C144, forming hemithioacetals in an (R)- or (S)-configuration. The differences in shape and the entrance of the substrate-binding pockets of CoV M^pro^s are mainly determined by three distinct structural features: (1) N or A residue in the S1 subsite, (2) α-helix/M residue or loop/T residue near the S2 subsite, (3) Q or P residue near the S4 subsite.

## Figures and Tables

**Figure 1 ijms-23-05669-f001:**
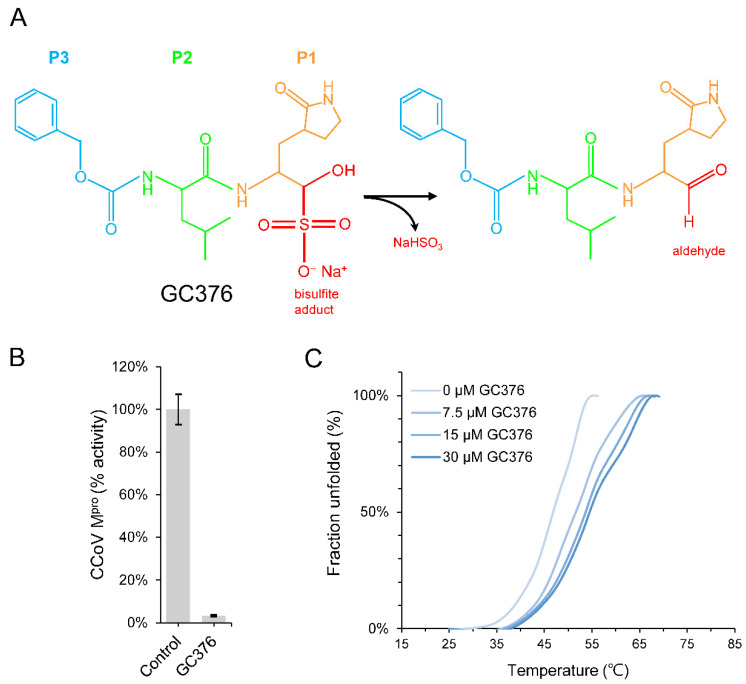
Functional characterization of the effects of GC376 on CCoV M^pro^: (**A**) The chemical structure of GC376 (left) and its aldehyde form (right); (**B**) in vitro enzyme activity assay of CCoV M^pro^ in the absence or presence of GC376; (**C**) dose-dependent stabilization effects of GC376 (0, 7.5, 15, and 30 μM) on the thermal stability of CCoV M^pro^.

**Figure 2 ijms-23-05669-f002:**
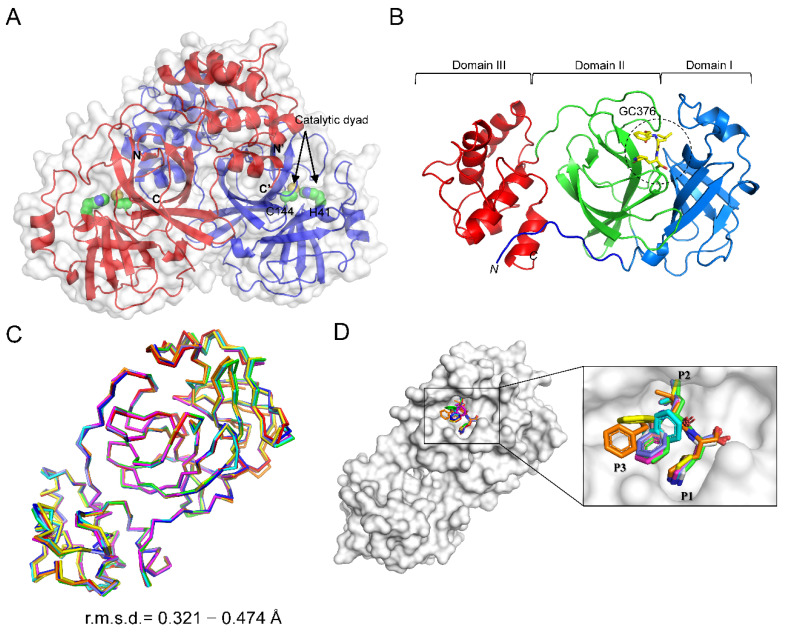
Overall structure of the CCoV M^pro^ in complex with GC376: (**A**) Dimeric assembly of CCoV M^pro^ (red and purple), the catalytic dyad (H41/C144) are shown as green spheres, the N-finger of one protomer extends into the substrate-binding pocket of the other protomer; the N- and C-terminus of each protomer are indicated; (**B**) domain organization of CCoV M^pro^: N-finger (residues 1–10 (blue)), domain I (residues 11–100 (marine)), domain II (residues 101–198 (green)), and domain III (residues 199–299 (red)), GC376 is shown as yellow sticks; (**C**) superimposition of the Cα backbone of the eight different protomers of CCoV M^pro^ in the same asymmetric unit; (**D**) comparison of the GC376 covalently linked to the C144 of eight different protomers of CCoV M^pro^ in the same asymmetric unit.

**Figure 3 ijms-23-05669-f003:**
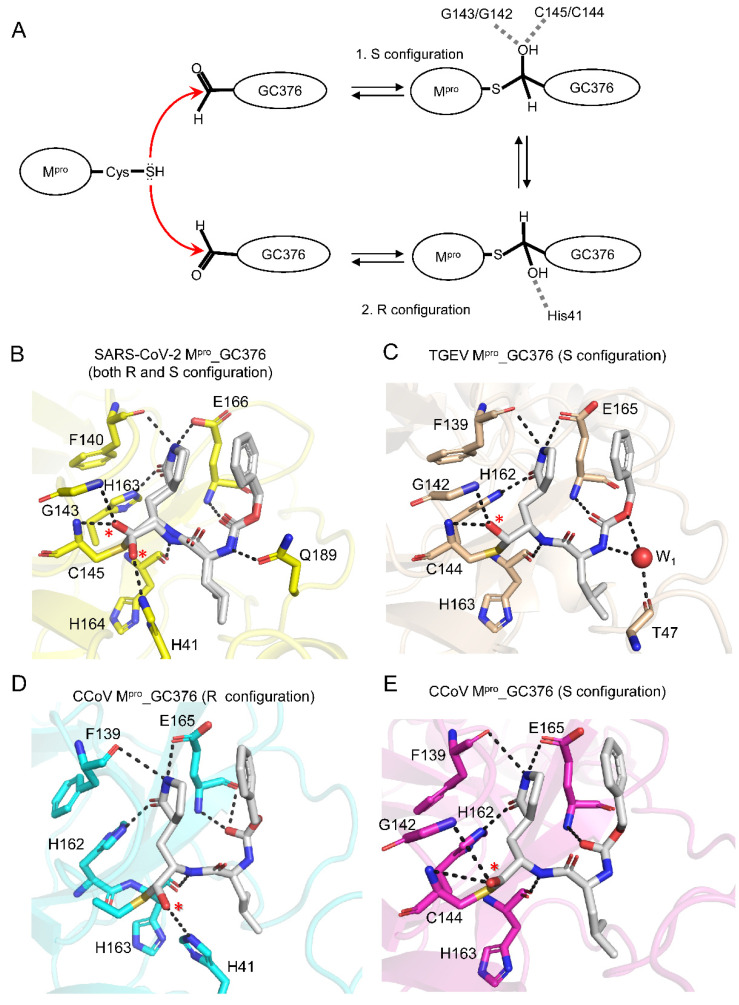
Comparison of the interactions between GC376 and different CoV M^pro^s: (**A**) Molecular mechanism underlying formation of an (R)- or (S)-configuration of GC376 by CoV M^pro^; (**B**–**E**) enlarged views of the substrate-binding pockets from (**B**) SARS-CoV-2 M^pro^_GC376 (PDB: 7CB7), (**C**) TGEV M^pro^_GC376 (PDB: 4F49), (**D**) protomer C of CCoV M^pro^_GC376, and (**E**) protomer A of CCoV M^pro^_GC376. The hydroxyl groups of the hemithioacetal from covalently linked GC376 are indicated by red star (*). H-bonds are shown as black dashed lines. A water molecule is shown as a red sphere.

**Figure 4 ijms-23-05669-f004:**
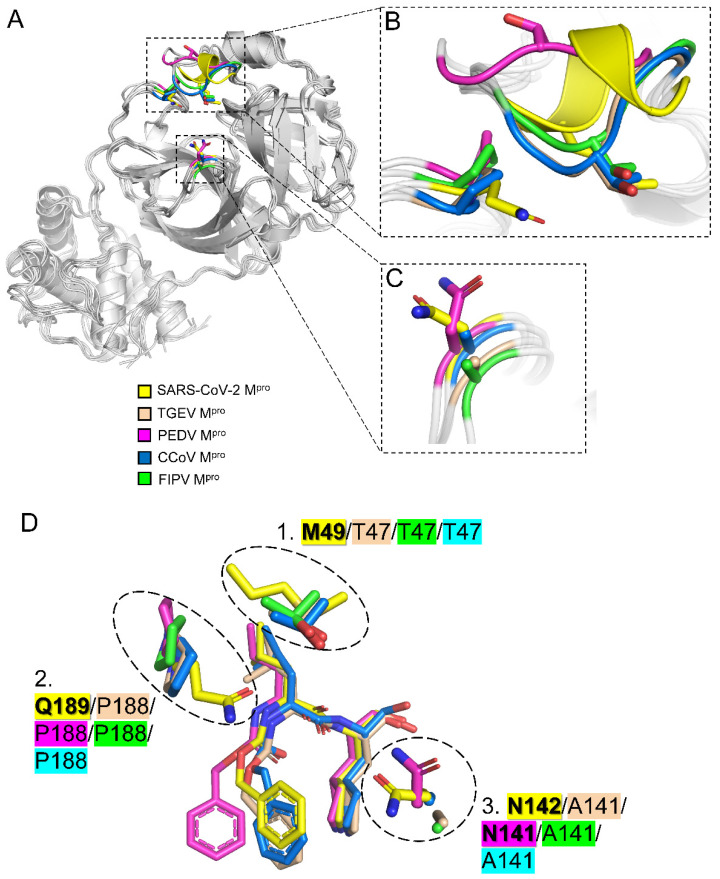
Structural comparison between animal CoV M^pro^s and SARS-CoV-2 M^pro^: (**A**) Overall structural comparison of SARS-CoV-2 M^pro^ (PDB: 7CB7), TGEV M^pro^ (PDB: 4F49), PEDV M^pro^ (PDB: 6L70), CCoV M^pro^, and FIPV M^pro^ (PDB: 5EU8). Three structural differences have been highlighted; (**B**) an enlarged view of the first (loop/α-helix) and second (P188/Q189) structural differences highlighted in (**A**); (**C**) an enlarged view of the third (A141/N142) structural differences highlighted in (**A**); (**D**) the three structural features of CoV M^pro^s that differentially contribute to recognition of GC376.

**Figure 5 ijms-23-05669-f005:**
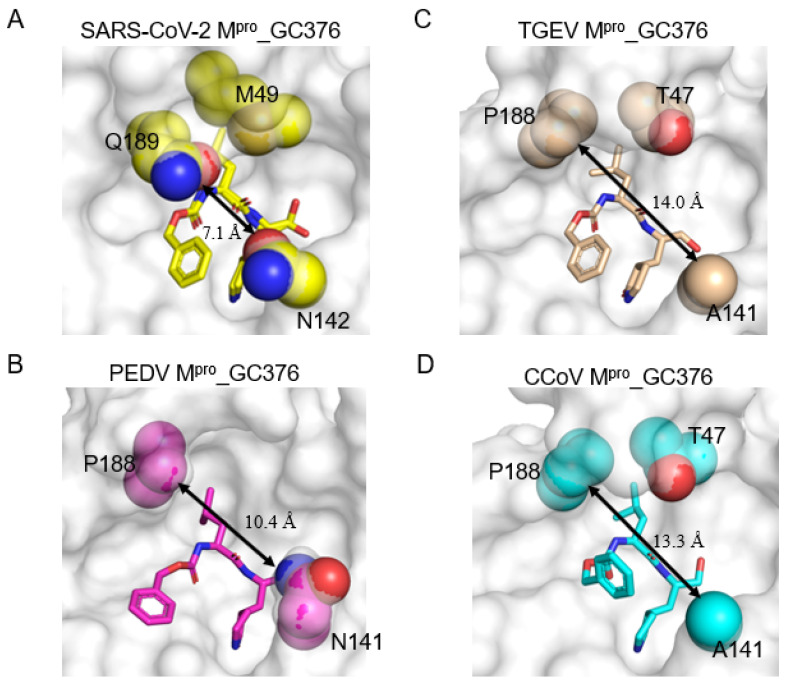
Comparison of the size of entrance of substrate-binding pocket among different CoV M^pro^s. Surface presentation of the substrate-binding pocket of: (**A**) SARS-CoV-2 M^pro^_GC376 (PDB: 7CB7, yellow); (**B**) PEDV M^pro^ (PDB: 6L70, magenta); (**C**) TGEV M^pro^_GC376 (PDB: 4F49, wheat); (**D**) CCoV M^pro^_GC376 (cyan). The three structural features demonstrated in Figure 4 within the substrate-binding pocket are shown in spheres as indicated. The shortest distances between sidechains of Q189/P188 and N142/(N/A)141 from SARS-CoV-2 M^pro^/animal CoV M^pro^s are measured and indicated by double-headed arrows. GC376s are shown in sticks.

**Figure 6 ijms-23-05669-f006:**
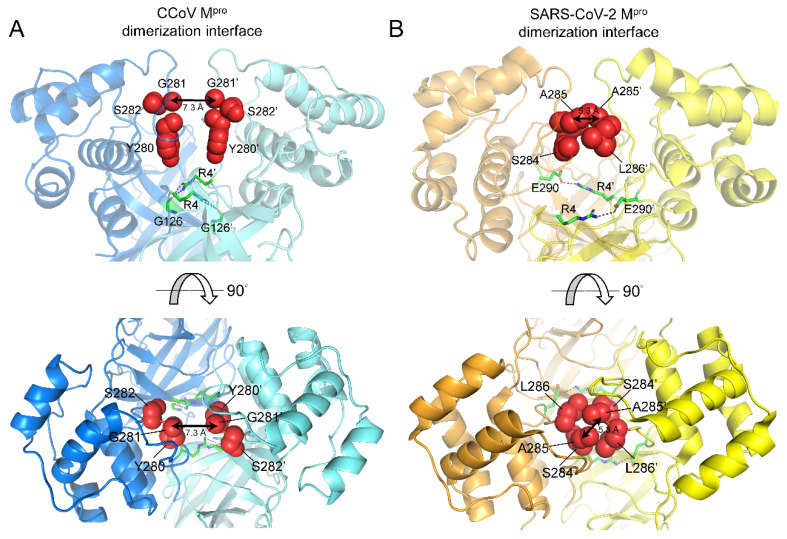
Comparison of the dimerization interfaces between (**A**) CCoV M^pro^ and (**B**) SARS-CoV-2 M^pro^. The three critical residues forming hydrophobic core at the dimerization interface of SARS-CoV-2 M^pro^ (S284-A285-L286) are shown in red spheres as compared with those of CCoV M^pro^ (Y280-G281-S282). The residues that participate in forming salt bridges (R4-E290 in SARS-CoV-2 M^pro^) and hydrogen bonding (R4-G126 in CCoV M^pro^) are shown as green sticks. The distances between the Cα atom of G281s or A285s are indicated.

**Table 1 ijms-23-05669-t001:** X-ray data collection and refinement statistics of GC376 bound CCoV M^pro^.

	GC376 Bound CCoV M^pro^
PDB Code	7XJW
Data collection	
Diffraction source	TPS 05A, 3 GeV TPS, NSRRC
Wavelength (Å)	0.99984
Detector	MX300-HS
Crystal-detector distance (mm)	300
Space group	*C*2
*a*, *b*, *c* (Å)	156.975, 125.749, 160.418
α, β, γ (°)	90, 97.467, 90
Resolution range (Å)	30.0–2.75 (2.85–2.75)
Total no. of reflections	295,387 (28,408)
No. of unique reflections	79,392 (7891)
Completeness (%)	99.3 (99.7)
Multiplicity	3.7 (3.6)
〈*I*/σ(*I*)〉	20.39 (3.39)
*R* _merge_	0.062 (0.386)
*R* _p.i.m._	0.036 (0.234)
CC_1/2_	(0.914)
Refinement	
Resolution range (Å)	27.83–2.75 (2.82–2.75)
Final *R*_work_ (%)	21.0 (27.2)
Final *R*_free_ (%)	25.7 (28.0)
No. of non-H atoms	18,564
No. of atoms	
Protein	18,224
Ligand	232
Water	108
*B* factors (Å^2^)	54.4
Protein	54.6
Ligand	48.6
Water	31.9
R.m.s. deviations	
Bonds (Å)	0.015
Angles (°)	1.72
Ramachandran plot	
Most favoured (%)	92.44
Allowed (%)	7.22
Outliers (%)	0.34

Values in parentheses are for the highest resolution shell.

## Data Availability

The data that support the findings of this study are available from the corresponding author upon reasonable request. The coordinates and structure factors of CCoV M^pro^ in complex with GC376 have been deposited in PDB with accession code 7XJW.

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
