# Peer review of "A Structural Comparison of SARS-CoV-2 Main Protease and Animal Coronaviral Main Protease Reveals Species-Specific Ligand Binding and Dimerization Mechanism"

_ijms, 2022, doi:10.3390/ijms23105669_

Round 1

Reviewer 1 Report

The authors of the manuscript reported the first crystal structure of main protease Mpro of canine coronavirus (CCoV) in complex with commercial compound GC376 which is known as a covalent inhibitor of many CoV Mpros. They also tested the interaction between the protein and GC376 using differential scanning fluorimetry (DSF) and enzymatic inhibition assay. The authors compared the structure of CcoV Mpro with different animal CoV Mpros including Sars CoV-2. Three important differences of amino acid residues in the active site of proteases were identified after aligning of multiple sequences of animal and human CoV Mpros and structural comparison of protein-ligand complexes.

The paper is appropriate for this journal and worth being published. 
Overall comments:

I was expecting the authors to determine the catalytic activity of CCoV Mpro. Similarly, Kd or IC50 values for CCov Mpro interaction with GC376 by DSF and inhibition assay could be estimated and compared with the published values for Sars CoV-2 Mpro and other animal CoV Mpros.

 I would suggest redrawing Figure 3A to clearly show the involvement of sulfur of cysteine145 in the formation of a covalent bond with the inhibitor (complex formation on the right side of fig3A).

Lines 164-166. Please check the descriptions of legends carefully.

The methods are not fully described. The purified protein storage buffer and the final quantity of the protein after purification should be mentioned. The CCov Mpro amino acid sequence contains additional amino acids at the N-terminus, and this should be clearly indicated in the methods section. It is known that Sars Cov-2 Mpro activity diminishes in the presence of additional amino acids at N-terminus (such as 6His). Can the authors predict how additional amino acids at the N-terminus in the CCov Mpro protein may affect protein activity?

5.2 section. The substrate concentration used in the enzymatic inhibition assay is not indicated.

5.3 section. DSF experiments were performed in Tris buffer; however, the pH of Tris buffer highly depend on the temperature (the pH at the protein Tm =46 oC will be about 7.2 instead of 7.8, since Tris temperature coefficient is of -0.026 pH unit per degree Celsius). The strength of the interaction (and thus the Tm shift) may depend on pH. Therefore, Hepes or phosphate buffer is a better choice for such experiments.

Author Response

Response to Reviewer 1 Comments

The authors of the manuscript reported the first crystal structure of main protease Mpro of canine coronavirus (CCoV) in complex with commercial compound GC376 which is known as a covalent inhibitor of many CoV Mpros. They also tested the interaction between the protein and GC376 using differential scanning fluorimetry (DSF) and enzymatic inhibition assay. The authors compared the structure of CcoV Mpro with different animal CoV Mpros including Sars CoV-2. Three important differences of amino acid residues in the active site of proteases were identified after aligning of multiple sequences of animal and human CoV Mpros and structural comparison of protein-ligand complexes.

The paper is appropriate for this journal and worth being published. 

Overall comments:

Point 1: I was expecting the authors to determine the catalytic activity of CCoV Mpro. Similarly, Kd or IC50 values for CCov Mpro interaction with GC376 by DSF and inhibition assay could be estimated and compared with the published values for Sars CoV-2 Mpro and other animal CoV Mpros.

Response 1: Yes, we originally plan to conduct the experiments. However, due to the rapidly increasing COVID-19 confirmed cases in Taiwan, our government implemented strict measures to isolate COVID-19 confirmed cases and potential contacts to control the pandemic. Many peoples were isolated at home and our works has been halted until the pandemic relief.

Point 2: I would suggest redrawing Figure 3A to clearly show the involvement of sulfur of cysteine145 in the formation of a covalent bond with the inhibitor (complex formation on the right side of fig3A).

Response 2: Thanks for your suggestion. The Figure 3A has been redrawn to clearly show the involvement of sulfur atom of Cys145 in the complex formation with GC376.

Point 3: Lines 164-166. Please check the descriptions of legends carefully.

Response 3: The descriptions of legends in lines 164-166 has been corrected.

Point 4: The methods are not fully described. The purified protein storage buffer and the final quantity of the protein after purification should be mentioned. The CCov Mpro amino acid sequence contains additional amino acids at the N-terminus, and this should be clearly indicated in the methods section. It is known that Sars Cov-2 Mpro activity diminishes in the presence of additional amino acids at N-terminus (such as 6His). Can the authors predict how additional amino acids at the N-terminus in the CCov Mpro protein may affect protein activity?

Response 4: Thanks for your suggestions. First, the storage buffer and the final quantity of the purified protein has been added. Second, the description that the purified CCoV Mpro contains an additional Gly residue at the N-terminus after TEV cleavage has been mentioned in the methods section. Third, the previous study has reported that the N-terminal extra Gly residue has caused about three-fold decrease on catalytic efficiency of SARS-CoV-2 Mpro compared with the native form. Interestingly, the first serine residues of all the solved CCoV Mpros can be clearly seen in the crystal structures and five of them still hydrogen bonded to both F139 and E165 to stabilize the formation of S1 subsites of CCoV Mpros. Therefore, we predicted that the extra Gly residue only slightly diminish the enzymatic activity of CCoV Mpro.

Point 5: 5.2 section. The substrate concentration used in the enzymatic inhibition assay is not indicated.

Response 5: The used substrate concentration for FRET-based assay has been added.

Point 6: 5.3 section. DSF experiments were performed in Tris buffer; however, the pH of Tris buffer highly depend on the temperature (the pH at the protein Tm =46 oC will be about 7.2 instead of 7.8, since Tris temperature coefficient is of -0.026 pH unit per degree Celsius). The strength of the interaction (and thus the Tm shift) may depend on pH. Therefore, Hepes or phosphate buffer is a better choice for such experiments.

Response 6: Thanks for your suggestions. Yes, in general, the HEPES or phosphate buffer is a better choice than Tris buffer in experiments with changing temperatures. However, in DSF experiments, the buffer system for different target proteins should be optimized to give the best results. In our case, the Tris buffer has the best stabilizing effect on CCoV Mpro and thus be chosen to perform the DSF experiments.

Reviewer 2 Report

In this study, Chien-Yi Ho et al. solved the crystal structure of the CCoV Mpro/GC376 complex. They analyzed the structural details and compared with that of several published CoV Mpro/GC376 structures. And they found species-specific substrate-binding and dimerization mechanism. Following are some suggestions.

Figure 1B. FRET was performed using a CCoV Mpro concentration of 9.4 uM, which seems very high. Published study probably using a concentration at nanomolar range. And the inhibitory effect of GC376 should be evaluated using a dose-dependent manner rather than using only one compound concentration (120 uM).

Figure 2D The GC376 of the different protomers showed structural difference especially at the P3 group. It would be good to show the density map of GC376 and also the main substrate binding residues from different protomers in the supporting information.

Figure S2 It seems a little bit difficult to read the residue colored in yellow, but it seems all yellow residues are “P”.

Line 313-314 What does epitopes for monoclonal antibody mean here? Mpro usually functions in the cytoplasm of the cells and antibodies in the plasma would not have access to this nonstructural protein.

Author Response

Response to Reviewer 2 Comments

In this study, Chien-Yi Ho et al. solved the crystal structure of the CCoV Mpro/GC376 complex. They analyzed the structural details and compared with that of several published CoV Mpro/GC376 structures. And they found species-specific substrate-binding and dimerization mechanism. Following are some suggestions.

Point 1: Figure 1B. FRET was performed using a CCoV Mpro concentration of 9.4 uM, which seems very high. Published study probably using a concentration at nanomolar range. And the inhibitory effect of GC376 should be evaluated using a dose-dependent manner rather than using only one compound concentration (120 uM).

Response 1: Thanks for your suggestions. Due to the rapidly increasing COVID-19 confirmed cases in Taiwan, our government implemented strict measures to isolate COVID-19 confirmed cases and potential contacts to control the pandemic. Many peoples were isolated at home and we could not perform these experiments until the pandemic relief.

Point 2: Figure 2D The GC376 of the different protomers showed structural difference especially at the P3 group. It would be good to show the density map of GC376 and also the main substrate binding residues from different protomers in the supporting information.

Response 2: The density map of GC376 and the residues involved in ligand binding has now been shown and added in the supporting information.

Point 3: Figure S2 It seems a little bit difficult to read the residue colored in yellow, but it seems all yellow residues are “P”.

Response 3: Yes, all the yellow residues are “P (proline)”. We have re-colored the Figure S2 for better reading.

Point 4: Line 313-314 What does epitopes for monoclonal antibody mean here? Mpro usually functions in the cytoplasm of the cells and antibodies in the plasma would not have access to this nonstructural protein.

Response 4: In addition to the rapid antigen tests and RT-PCR method to detect the presence of SARS-CoV-2, we also need the antibodies test to confirm the production of specific antibodies in human bodies against SARS-CoV-2 after natural infections or vaccination. Currently, most of the serological tests detect the antibodies against spike (S) protein or nucleocapsid (N) protein. It has been reported that IgG Abs specific for SARS-CoV-2 Mpro has been detected in serum and saliva of COVID-19 patients [J Immunol December 1, 2020, 205 (11) 3130-3140]. Therefore, the structural differences of different coronaviral Mpros can be utilized to develop specific Abs to detect the infections of different coronaviruses to avoid cross-reactivity.

Round 2

Reviewer 2 Report

No more questions.